# Comparison of Pneumonia Severity Indices, qCSI, 4C-Mortality Score and qSOFA in Predicting Mortality in Hospitalized Patients with COVID-19 Pneumonia

**DOI:** 10.3390/jpm12050801

**Published:** 2022-05-16

**Authors:** Isil Kibar Akilli, Muge Bilge, Arife Uslu Guz, Ramazan Korkusuz, Esra Canbolat Unlu, Kadriye Kart Yasar

**Affiliations:** 1Department of Pulmonary Disease, Bakirkoy Dr. Sadi Konuk Training and Research Hospital, University of Health Sciences, Dr. Tevfik Saglam Street, No. 11, Bakirkoy, Istanbul 34147, Turkey; 2Department of Internal Medicine, Bakirkoy Dr. Sadi Konuk Training and Research Hospital, University of Health Sciences, Dr. Tevfik Saglam Street, No. 11, Bakirkoy, Istanbul 34147, Turkey; mugebilge@yahoo.com; 3Department of Pulmonary Disease, Mehmet Akif Ersoy Training and Research Hospital, University of Health Sciences, Turgut Ozal Boulevard, No. 11, Kucukcekmece, Istanbul 34303, Turkey; arifegz@yahoo.com; 4Department of Infectious Disease, Bakirkoy Dr. Sadi Konuk Training and Research Hospital, University of Health Sciences, Dr. Tevfik Saglam Street, No. 11, Bakirkoy, Istanbul 34147, Turkey; ramazankorkusuz@hotmail.com (R.K.); canbolat-esra@hotmail.com (E.C.U.); hkkyasar@gmail.com (K.K.Y.)

**Keywords:** COVID-19 pneumonia, prediction, prognosis, severity, mortality, risk scores

## Abstract

This is a retrospective and observational study on 1511 patients with SARS-CoV-2, who were diagnosed with COVID-19 by real-time PCR testing and hospitalized due to COVID-19 pneumonia. 1511 patients, 879 male (58.17%) and 632 female (41.83%) with a mean age of 60.1 ± 14.7 were included in the study. Survivors and non-survivors groups were statistically compared with respect to survival, discharge, ICU admission and in-hospital death. Although gender was not statistically significant different between two groups, 80 (60.15%) of the patients who died were male. Mean age was 72.8 ± 11.8 in non-survivors vs. 59.9 ± 14.7 in survivors (*p* < 0.001). Overall in-hospital mortality was found to be 8.8% (133/1511 cases), and overall ICU admission was 10.85% (164/1511 cases). The PSI/PORT score of the non-survivors group was higher than that of the survivors group (144.38 ± 28.64 versus 67.17 ± 25.63, *p* < 0.001). The PSI/PORT yielding the highest performance was the best predictor for in-hospital mortality, since it incorporates the factors as advanced age and comorbidity (AUROC 0.971; % 95 CI 0.961–0.981). The use of A-DROP may also be preferred as an easier alternative to PSI/PORT, which is a time-consuming evaluation although it is more comprehensive.

## 1. Introduction

Severe acute respiratory syndrome Coronavirus 2 (SARS-CoV-2), known as the “COVID-19 pandemic”, is globally associated with high mortality. SARS-CoV-2 has infected over 364,191,457 million people in the world, causing over 5,631,457 million casualties [1]. COVID-19 usually exhibits mild or moderate (81%) clinical appearance, even though 14% of cases are severe and 5% are critical. According to the Chinese Center for Disease Control and Prevention report, including 72,314 COVID-19 cases, the overall mortality rate was reported as 2.3%. This rate was 14.8% for patients older than 80 years, and 49% in critical cases [2].

COVID-19 is a multisystemic viral disease which may show varying clinical presentations such as asymptomatic disorders, injuries associated with organs as liver, heart, kidney, neurological manifestatations, coagulopathy, sepsis, septic shock and multiple organ dysfunction [3,4,5]. Among the above mentioned problems, especially acute respiratory failure, adult respiratory distress syndrome (ARDS) and/or multiorgan failure are known as the major complications of COVID-19, leading to death. In critical cases, rare complications such as cytokine storm and macrophage activation syndrome may also be observed [6,7].

As our knowledge of COVID-19 has evolved, certain clinical predictors leading to poor prognosis have been identified. These well-established risk factors for severe COVID-19 are: comorbidites related to old age, hypertension, cardiovascular diseases, diabetes, obesity, chronic lung disease (especially chronic obstructive and interstitial lung diseases), immunocompromised state, end-stage renal disease, liver disease and malignancy [8,9,10]. It is also reported that lymphopenia, increased levels of ferritin, d-dimer, troponin I, lactate dehydrogenase (LDH) and intereukin-6 were associated with poor prognosis and higher mortality [7,8,11,12,13,14]. 

As there is still an uncertainty about the progress of COVID-19 disease, it becomes increasingly important to develop clinical risk classification tools to identify the patients under risk, to prognose their clinical progress and to predict their mortality rate [15]. Common prognostic scales, widely used in community-acquired pneumonia (CAP) and sepsis as proven disease severity scoring systems in order to predict the disease consequences, are also referred to in COVID-19 [16,17,18]. On the other hand, novel models specifically designed for COVID-19 are also being developed [19,20,21,22,23].

Previously published studies did not show differentiating factors among patients with MERS and those without MERS [24]. On the other hand, Rainer et al. demonstrated that scoring systems might help identify patients who should receive more specific tests for influenza or SARS [25]. Nowadays, CAP severity indices which are known to be significantly associated with mortality are examined almost in all pneumonia cases. As an example, a multi-center prospective study showed that during a H1N1 virus pandemic severity of pneumonia was identified by PSI/PORT score [26]. When the utility of severity indices in viral pneumonia were examined, PSI/PORT was an important indicator for assessing the prognosis of patients suffering from CAP in which the causative agent to be a respiratory virus or not [27]. Several studies showed that CURB-65 score was a useful tool in influenza, non-influenza, bacterial and mixed viral-bacterial agents in CAP [28,29].

Pneumonia Severity Index (PSI/PORT) [30], A-DROP (Age, dehydration, oxygen saturation, orientation, blood pressure) [31], National Early Warning Score 2 (NEWS2) [32], Modified Early Warning Score (MEWS) [33], CURB-65 (confusion, urea, respiratory rate, blood pressure, age) [34], expanded CURB-65 (hypoalbuminemia, LDH, thrombocytopenia, confusion, urea, respiratory rate, blood pressure, age) [35] and the quick Sequential (Sepsis-Related) Organ Failure Assessment (qSOFA) [36] are some of the other scoring systems used in COVID-19 cases as well. However, their performance on prognosis and mortality prediction is not presented in sufficent detail [17,37,38,39,40,41,42,43].

The quick COVID-19 Severity Index (qCSI: mental status, respiratory rate and systolic blood pressure) [19] and ISARIC 4C Mortality score (4C Mortality score) [20] are specifically developed for COVID-19 as novel risk assessment tools.

Scoring systems can be used for early identification of high-risk patients, accurate assessment of disease severity, prediction of disease progress and determination of patient specific treatment approach [16,18,44,45].

In this retrospective study, we aimed to identify simple, useful and accurate scoring systems by studying the conventional methods used for community acquired pneumonia (CAP) and sepsis, and those specifically used for COVID-19 such as qCSI and 4C mortality.

## 2. Materials and Methods

### 2.1. Study Design and Participants

This retrospective observational study was carried out on 1909 patients with SARS-CoV-2, who were diagnosed with COVID-19 pneumonia by real-time PCR testing and hospitalized at Prof. Dr. Murat Dilmener Emergency Hospital, pandemic 3rd level, in Istanbul, from 1 September 2020 to 31 December 2020. The diagnosis of COVID-19 was determined on the basis of World Health Organization (WHO) guidelines [46]. All cases enrolled in the study were over the age of 18, and none of them had been taken to the intensive care unit (ICU). They were managed in accordance with the COVID-19 treatment protocol of the Turkish Health Ministry [47]. The research was approved by the ethics committee of the University of Health Sciences, Bakırköy Dr. Sadi Konuk Training and Research Hospital (approval number 2021/91), and was conducted following the principles of the Declaration of Helsinki.

Retrospectively, medical records were obtained from the hospital electronic database. All the parameters relevant for scoring the disease severity were measured and recorded. A standardized form was used for data collection, which included demographics, past medical history, underlying chronic diseases, vital signs, the severity of admission, laboratory findings and chest computed tomography (CT) scans results. In addition, definitive outcomes (death, discharge or ICU admission) were obtained from the hospital information system.

Patients with COVID-19 were categorized into two groups (non-severe and severe illness) according to National Institutes of Health (NIH) classification based on disease severity [48]. Severe cases were defined as having at least any one of the following criteria: respiratory distress (>30 breaths/min); oxygen saturation < 94% at rest; arterial partial pressure of oxygen (PaO_2_)/fraction of inspired oxygen (FiO_2_ < 300 mmHg), or lung infiltrates > 50%. Radiologist-evaluated chest CT scans were classifed into three categories, mild, moderate and severe involvement [49]. Patients with no radiological involvement were excluded in our study. Only moderate and severe cases were included in our study. Patients who had incomplete data or those who had pneumonia arising from other pathogens were all excluded from the study. Another criterion for exclusion was pregnancy. After the exclusion, 1511 patients (older than 18 years) who were hospitalized with the diagnosis of COVID-19 pneumonia were included in the study. The patient flowchart can be seen in Figure 1.

### 2.2. Scores Selections and Definitions

The scores on hospitalization of each patient were calculated for nine severity scoring rules, including PSI/PORT, A-DROP, NEWS-2, MEWS, CURB-65, Expanded CURB-65, qSOFA, qCSI, and 4C Mortality as defined previously in the literature [19,20,30,31,32,33,34,35,36]. All the components of each scoring system were accurately registered in the medical records.

### 2.3. Statistical Analysis

All statistical analyses were performed in commercially available SPSS software v.22 (Statistical Package for the Social Sciences Inc., Chicago, IL, USA). Patient characteristics were summarized using descriptive statistics (mean, standard deviation, median, frequency, percentages, minimum, maximum, Q1–Q3). The conformity of the quantitative variables to the normal distribution was tested with the Shapiro-Wilk test and graphical examinations. Student *t*-test was used for the comparison of normally distributed quantitative variables between two groups, and the Mann-Whitney U test was used for comparisons between two groups of non-normally distributed quantitative variables. Categorical variables were compared using the Chi-square test and Fisher’s exact test. Diagnostic screening tests (sensitivity, specificity, positive predictive value and negative predictive value) and ROC (area under the curve of the receiver operating characteristic) analysis were used to determine the cutoff value for all nine scoring systems. ROC Curve analysis and Binomial exact test were used to determine in-hospital mortality prediction. Multivariate logistic regression analysis was performed to identify other risk factors on mortality. A *p*-value < 0.05 was accepted as statistically significant.

## 3. Results

### 3.1. Comparison of Basic Clinical Characteristics between the Two Groups

1511 patients, 879 male (58.17%) and 632 female (41.83%) with a mean age of 60.1 ± 14.7, were included in the study. Although gender difference was not statistically significant, 80 (60.15%) of the patients who died were male. Mean age was 72.8 ± 11.8 in non-survivors vs. 59.9 ± 14.7 in survivors (*p* < 0.001). 386 patients (25.54%) had only a single comorbidity, while 693 (45.86%) had two or more. The most common comorbidity was hypertension (48.04%, 726/1511), followed by diabetes mellitus (33.28%, 503/1511) and coronary artery disease (14.36%, 217/1511). The number of comorbidities was higher in patients who died (78.19% vs. 42.74%, *p* < 0.001). Hypertension, coronary artery disease, atrial fibrillation (AF), congestive heart failure and cerebrovascular disease were significantly more common in non-survivors than in survivors (*p* < 0.001). We observed a difference between cohorts in terms of malignancy and chronic kidney diasease (*p* = 0.01), while no such difference existed for diabetes, dyslipidemia, COPD and asthma. Ultimately, the non-survivor group was older and had more comorbidities than the survivor group (*p* < 0.001 for both). Respiratory rate, oxygen saturation by pulse oximetry (SpO_2_) under oxygen support, supplemental oxygen requirement and heart rate were significantly higher in non-survivors.

General information and baseline characteristics of the patients are shown in Table 1. Laboratory findings, CT scores, disease severity status and outcomes of the patients are given in Table 2.

### 3.2. Comparison of Laboratory Tests and Chest CT Scans between the Two Groups

Patients had decreased lymphocyte, platelet, albumine and elevated neutrophil count, N/L ratio, C-reactive protein (CRP), urea, creatinine, LDH, ferritin, D-dimer, INR, troponin I levels in non-survivors (*p* < 0.001). In addition to that, glucose, ALT, AST and fibrinogen levels were higher in non-survivors (*p* values, respectively 0.003, 0.01, 0.04 and 0.01). There were no significant differences in procalcitonin and hematocrit levels between two groups. Concurrently, deceased patients had significantly higher disease severity status and chest CT scores than the survivors (*p* < 0.001). Additionally, 97.75% (130/133) of deceased patients had severe disease status.

### 3.3. Score Distribution

The PSI/PORT score of the non-survivor group was higher than that of the survivor group (144.38 ± 28.64 versus 67.17 ± 25.63, *p* < 0.001). Similarly, all the other eight scores were also found to be higher in non-survivors than survivors (*p* < 0.001). As seen in Table 3, prognostic scores were higher in deceased patients.

When the mortality rates were investigated in terms of pneumonia scores, the mortality rate in the high-risk group was found to be 6.88% (104 cases) for those having a NEWS2 score of 7 or higher. The mortality rate reduced to 6.22% (94 cases) for patients having Expanded CURB-65 scores of 4 or higher, and to 5.69% (86 cases) for those classified as PSI/PORT Class V. When the same analysis was performed in low-risk groups, there was no mortality among all 872 patients categorized as Class II with regard to PSI/PORT, and for patients having 4C Mortality score between 0 and 3, only a single mortality was observed.

ICU admissions were 8% (122 cases) in NEWS2 (≥7 points), 5.75% (87 cases) in PSI/PORT Class V and 0.13% (2 cases) in 4C Mortality (0–3 points). Prediction scores distributions for mortality and ICU admission are given in Table 4.

Among all nine scores that PSI/PORT presented the highest discrimination (AUROC 0.971; 95% CI 0.961–0.981), followed by A-DROP (AUROC 0.929; 95% CI 0.911–0.948), NEWS2 (AUROC 0.885; 95% CI 0.860–0.909), qCSI (AUROC 0.882; 95% CI 0.853–0.911), 4C-Mortality (AUROC 0.875; 95% CI 0.845–0.906), MEWS (AUROC 0.870; 95% CI 0.842–0.898), CURB-65 (AUROC 0.859; 95% CI 0.823–0.896), Expanded CURB-65 (AUROC 0.836; 95% CI 0.800–0.873), and qSOFA (AUROC 0.818; 95% CI 0.786–0.850) in predicting in-hospital death (Table 5, Figure 2). Overall, PSI/PORT score showed higher sensitivity and specificity for in-hospital mortality than the other scores. When the optimal cut-off value of PSI/PORT was taken as 107, the sensitivity and specificity were obtained as 91.7% and 91.9%, respectively. The cut-off value for the A-DROP score, which turned out to be the second best result, was assumed to be 2 yielded a sensitivity and specificity of 84.2% and 86.1%, respectively (Table 5).

Binomial exact test has figured out PSI/PORT in predicting mortality as a superior reference, compared with A-DROP, NEWS2, qCSI, 4C-Mortality, MEWS, CURB-65, expanded CURB-65 and qSOFA (*p* < 0.001). Similarly taking A-DROP as a reference, A-DROP was superior compared with NEWS2 (*p* = 0.002), qCSI (*p* = 0.005), 4C-Mortality (*p* < 0.001), MEWS (*p* < 0.001), CURB-65 (*p* < 0.001), expanded CURB-65 (*p* < 0.001) and qSOFA (*p* < 0.001). There was no statistically significant difference between CURB-65, expanded CURB-65 and qSOFA (*p* > 0.005). 

### 3.4. Outcomes in Two Groups

In our study, in-hospital mortality was found to be 8.8% (133/1511 cases), and overall ICU admission was 10.85% (164/1511 cases). While the total number of patients who died in ward and ICU was found to be 133, the remainder (1378) were discharged. 114 patients died in ICU, and 19 died while they were being treated in the ward. It was found that the mean hospitalization stay of the deceased patients was longer (11.27 vs. 14.52 days, *p* < 0.001), and the frequency of admission to the ICU was higher (85.71% vs. 3.62%, *p* < 0.001).

Mortality was associated with advanced age, presence of certain comorbidities (hypertension, coronary artery disease, AF, congestive heart failure, cerebrovascular disease), hypoxia or tachypnea on admission, higher urea, creatinine, D-dimer, troponin I, ferritine, CRP, neutrophile count, neutrophil-to-lymphocyte ratio, lower lymphocyte count, platelet count and albumine on admission.

Backward stepwise logistic regression test was performed on 32 parameters predicting mortality including age, number of comorbidities, presence of certain comorbidities (hypertension, coronary artery disease, AF, congestive heart failure, cerebrovascular disease, chronic kidney disease, malignancy), SpO_2_, O_2_ support, heart rate, neutrophil-to-lymphocyte ratio, platelet count, troponin I, D-dimer, INR, fibrinogen, ferritin, CRP, glucose, urea, creatinine, albumine, AST, ALT, LDH, disease severity status, CT score, hospital length of stay, PSI/PORT, A-DROP, NEWS2, qCSI, 4C-Mortality, MEWS, CURB-65, expanded CURB-65 and qSOFA; and the independent risk factors for mortality came out to be PSI/PORT, A-DROP, MEWS, qSOFA scores, O_2_ support, PLT and CRP ve LDH (Table 6). 

In this model, PSI/PORT score ≥ 107 predicts a 25.172 times mortality (%95 CI: 11.232–56.413). A-DROP score ≥ 2 predicts a 4.686 times higher mortality (%95 CI: 2.303–9.523). MEWS score ≥ 3 predicts a mortality risk 2.458 times higher (%95 CI: 1.255–4.814). qSOFA score ≥ 1 predicts a 5.714 times higher mortality (%95 CI: 1.774–18.399). One unit lower O_2_ support predicts 1.065 times higher mortality (%95 CI: 1.027–1.105). Thrombocytopenia increases mortality risk by 0.997 (%95 CI: 0.995–1.000). One mg/dl CRP rise increases mortality risk by 0.996 times (%95 CI: 0.992–1.000). One unit increase in LDH increases mortality risk by 1.003 times (%95 CI: 1.001–1.004) (Table 6). PSI/PORT, A-DROP, MEWS, qSOFA scores, O_2_ support, PLT, CRP and LDH came out as independent risk factors predicting in-hospital mortality.

## 4. Discussion

We aimed to analyze the utility of the well-known CAP severity indices; CURB-65, Expanded CURB-65, PSI-PORT, NEWS2, MEWS and A-DROP, as well as qCSI 4-C Mortality and qSOFA scores introduced for COVID-19 in predicting mortality and progression to severe disease.

In our study, in-hospital mortality was found to be as 8.8% (133 cases). This result seems to be lower than those in other studies though it stands as a higher value when compared with 0.77% (cases: 11,249,216, deaths: 86,661), which comes from estimations based on national data [1,8,50].

In our study, mortality was associated with factors such as; advanced age presence of certain comorbidities (hypertension, coronary artery disease, AF, congestive heart failure, cerebrovascular disease), hypoxia or tachypnea on admission, heart rate, higher urea, creatinine, D-dimer, troponin I, ferritine, CRP, neutrophile count, neutrophil-to-lymphocyte ratio, lower lymphocyte count, platelet count and albumine on admission similar to previous studies [4,8,51,52,53,54].

Multivariable logistic regression model revealed PSI/PORT, A-DROP, MEWS, qSOFA scores, O_2_ support, PLT, CRP and LDH as independent predictors for mortality. This may be the result of SARS-Cov-2 infection associated with hypoxia, thrombogenesis, inflammation and organ injury in concordance with previous studies [8,13,14,51,55,56].

Although no relationship was found between diabetes and mortality, hyperglycemia was found to be more frequent in those patients who died. Similarly, the disease severity and chest CT scores were found to be related with mortality [8,52,53,57,58,59].

The higher score at admission have higher risk of ICU care and death in patients with COVID-19 pneumonia. High prognostic scores indicate worse prognosis. All prognostic scores were higher in deceased patients. The mortality in low-risk groups that were designated to manage outside the hospital was 0 in PSI/PORT Class II, 1.73% (21/1208) in A-DROP low risk and 2.33% (26/1114) in CURB-65 low-risk.

When the high-risk groups were investigated the mortality rate was observed to be as 17.35% (42/242) in PSI/PORT Class IV and 74.78% (86/115) in PSI/PORT Class V. 30-day mortality in CAP patients in PSI Class and Mortality in the Pneumonia PORT Validation Cohort group were reported to be 0.6%, 9.3% and 27% for class II, IV and V patients, respectively [30].

All nine scoring systems evaluated in this study performed well in predicting in-hospital mortality. However, PSI/PORT (AUROC 0.971; 95% CI 0,961–0,981) had the highest predicting power, followed by A-DROP (AUROC 0.929; 95% CI 0.911–0.948). PSI/PORT score had both the highest sensitivity (94%) and the specificity (90%). Our results yielded an AUROC value of 0.971 for PSI-PORT, which was higher than those obtained in other COVID-19 studies with the results of AUROC values of 0.85 (95% CI 0.81–0.88) [16], 0.85 (95% CI 0.78–0.90) [37] and 0.83 (95% CI 0.82–0.84) [60]. While expanded CURB-65 had 0.836 AUROC (95% CI 0.800–0.873) and 82.3% specificity but with the lowest sensitivity of 70.7%, qSOFA had 0.818 AUROC (95% CI 0.786–0.850) and 97% sensitivity, but with the lowest specificity of 54.2% (Table 5).

Previous studies have already shown that PSI/PORT score was demonstrated to have a strong predictive performance in CAP cases. We have obtained even a higher performance on our COVID-19 pneumonia cohort than those found on the CAP, in which PSI/PORT presented an AUROC of 0.78 (95% CI 0.73–0.82) to predict composite primary endpoint (death within 28 days by any cause, or transfer to ICU) [61] and 0.812 (95% CI 0.673–0.951) to predict 30-day mortality [62].

PSI/PORT score consists of certain parameters such as age, male gender, comorbidities, metabolic abnormalities, tachypneia and hypoxemia which are shown to have a relation with mortality in COVID-19 patients [3,4,8,9,18,19,51,63,64]. The higher performance of PSI/PORT score may be attributed to the fact that the mean age of non-survivors was 72.8 ± 11.8, or that 78.19% (104/133) of them had two or more comorbidities and lower SpO_2_ levels compared with survivors. PSI/PORT score may be the best predictor for mortality, since it gives extra credit for such factors as advanced age, clustered comorbidities and hypoxemia. However, PSI/PORT may have some disadvantages as not to include chronic lung diseases as COPD and asthma; only evaluating heart failure in cardiologic parameters; limiting the respiratory rate above 30 per minute; underestimating severe pneumonia in young healthy patients due to absolute age parameters; automatic classification of individuals without any comorbidities over age 50 as Class II; and the fact that clinicians may find considering 19 parameters for score quantification more time-consuming.

Previous studies have shown that the A-DROP scoring system was accurate and clinically useful for assessing the severity of both bacterial and atypical pneumonia [65,66]. Recently, Miyashita et al. reported that a high A-DROP score, indicating severe or extremely severe pneumonia, was associated with a high mechanical ventilation rate or high death rate [67].

A-DROP score is a score which is calculated by adding such parameters as advanced age (male ≥ 70 years, female ≥ 75 years versus > 65 years in CURB-65), and respiratory failure (arterial oxygen saturation ≤ 90% or arterial oxygen pressure ≤ 60 mmHg) parameters to CURB-65. A-DROP score had the highest second rank in predicting in-hospital mortality with 0.929 AUROC (95% CI 0.911–0.948). In contradiction, there are also studies reporting that A-DROP score performs better than the PSI/PORT in predicting the in-hospital mortality of COVID-19 patients AUROCs for A-DROP 0.87 (95% CI 0.84–0.90) vs. PSI 0.85 (95% CI 0.81–0.88) [16] and AUROCs for A-DROP 0.875 (95% CI; 0.822–0.937) vs. PSI 0.873 (95% CI 0.820–0.925) [68]. These findings suggest that involving advanced age [4,8,18], which is related to higher COVID-19 mortality, places A-DROP score in a superior situation compared to CURB-65 in predicting mortality. In our study, the mean age of deceased patients was 72 years, and they had lower SpO_2_ levels. Zhou et al. also reported that the mean age of non-survivors with COVID-19 was 69 years [8,18] and **Xie** et al. reported hypoxemia was independently associated with in-hospital mortality.

In our study, the capability of PSI /PORT (AUROC 0.971) and A-DROP (AUROC 0.929) to predict hospital mortality was better than other studies in COVID-19 pneumonia, with the results of AUROC PSI/PORT values of 0.85 (95% CI 0.81–0.88) [16], 0.85 (95% CI 0.78–0.90) [37], 0.835 (95% CI 0.826–0.845) [60], 0.874 (95% CI 0.808–0.939) [18], 0.873(95% CI 0.820–0.925) [68], while AUROC A-DROP results were 0.87 (95% CI 0.84–0.90) [16] and 0.875 (95% CI 0.822–0.937) [68].

NEWS2 score, having the respiratory parameters (respiratory rate, oxygen saturation, supplemental oxygen), performed as the third-highest system in predicting mortality. Its lack of a scale to indicate increased oxygen requirement, its insensitivity to hypoxic respiratory failure (type 1) often encountered in COVID-19 cases and its seldom monitoring of hypercapnic respiratory failure (type 2), neglecting the age, comorbidities and organ dysfunctioning, come forward as the disadvantages of NEWS2 [7,11,32,50,69,70].

In our study NEWS2 ≥ 7 also predicted in-hospital mortality yielding the AUROC 0.885 (95% CI 0.860–0.909) which turned out to be better than those in the previous studies with AUROC 0.81 (95% CI 0.77–0.85) [16], 0.809 (95% CI 0.727–0.891) [17], 0.822 (0.690–0.953) [43].

qCSI score index, which includes only three respiratory parameters, was found to have a higher predictive performance with AUROC value 0.882 (95% CI 0.853–0.911) compared with Navaa et al., and Covino et al. AUROC 0.711 (95% CI 0.656–761) and AUROC 0.749 (95% CI 0.685–0.806), respectively [23,71]. Haimovich et al. had a AUROC 0.81 (95% CI 0.73–0.89) in predicting acute respiratory failure in accordance with our findings [19].

4C Mortality which was specifically developed for COVID-19 cases, came out to be the fifth rank in our score performance results, although it included parameters as age, inflammatory markers and respiratory measures yet underestimating comorbidities dealing them only by number. 4C Mortality score with AUROC 0.78 (95% CI 0.75–0.81) was reported in the second place after the PSI/PORT score having 0.79 (95% CI 0.77–0.82) AUROC in performance in a recent study [72]. 4C Mortality was found to have a good prognostic value for mortality on patients over the age of 60 (AUROC 0.799; 95% CI 0.738–0.851) and qCSI had a similar effectiveness as reported in (AUROC 0.749; 95% CI 0.685–0.806) [71].

qSOFA (AUROC 0.818; 95% CI 0.786–0.850), CURB-65 (AUROC 0.859; 95% CI 0.823–0.896), Expanded CURB-65 (AUROC 0.836; 95% CI 0.800–0.873) and MEWS (AUROC 0.870; 95% CI 0.842–0.898) scores were inferior in predicting in-hospital mortality of patients with COVID-19 pneumonia, as confirmed by previous studies with the results of AUROC for qSOFA 0.73 (95% CI 0.69–0.78), CURB-65 0.85 (95% CI 0.81–0.89) [16], qSOFA 0.63 (95% CI 0.6–0.66), CURB 65 0.74 (95% CI 0.72–0.77) [72], MEWS 0.670 (95% CI 0.573–0.767) [17], MEWS 0.586 (95% CI 0.531–0.640), qSOFA 0.673 (95% CI 0.620–0.723) [73], expanded CURB-65 0.885 (95% CI 0.827–0.942) [74].

Kodama et al. reported that expanded CURB-65 score was a good predictor, with 0.832 (95% CI 0.763–0.901) AUROC of an increase in oxygen requirement patients with SARS-CoV-2 pneumonia [75].

Fan et al. has also compared in-hospital mortality of different risk scores systems like our study [16]. Although both studies took A-DROP, CURB-65 and 4C Mortality scores by the same cut off values; we have used a different cut off value as 107 points for PSI/PORT score, which is a mid-value between Class III and IV patients. Fan et al. has found the greatest score as A-DROP; our results suggest that a cut-off value of 107 points with PSI/PORT score yield a more valid score for predicting in-hospital mortality.

In our study, in-hospital mortality was found to be as 8.8% (133 cases). As it was reported to WHO, that there have been 11,249,216 confirmed COVID-19 cases, of which 86.661 were fatal (0.77%) in Turkey during the time interval from January 2020 to January 2022 [1,8,50]. It is suggested that the relative highness of our mortality value, in comparison with the national statistics can be attributed to our cohorts, consisting of moderate and severe cases with an average age of 60.12 ± 14.73. An additional factor might be that the study was conducted in a hospital dedicated entirely to COVID-19 patients. Varying mortality rates were reported in other studies ranging from 2.3% to 36%. The characteristics of the patients included in these studies, different treatment regimens applied and different mortality criteria (e.g., in-hospital, 48 h, 72 h, 7-day or 30-day measures) used might have been the major reasons why they yielded such a wide range of different mortality rates [2,69,71,76].

### Limitation

This study has some limitations. It is a single-center study involving only hospitalized patients with moderate and severe disease in a retrospective design. Omicron and delta variants were not included, since the study was conducted during the early stages of the COVID-19 pandemic.

## 5. Conclusions

In conclusion, the accuracies of a variety of severity scores to predict in-hospital mortality in COVID-19 pneumonia patients were examined in our study. We found that PSI/PORT score showed the highest accuracy of in-hospital death prediction compared to other widely used CAP-specific and COVID specific score systems, such as qCSI and 4C Mortality. The PSI/PORT with a cut-off value of 107 points yielding the greatest performance was the best predictor for mortality, since it incorporated the factors such as advanced age and comorbidities. On the other hand, determination of the PSI/PORT required a longer time, since it involved the measurement and evaluation of a wide range of clinical parameters. Although all these data have to be validated in newer patient groups involving Cmicron and delta variants; our findings suggest that the use of A-DROP may also be preferred as a practical alternative to PSI/PORT, which is more time-consuming.

## Figures and Tables

**Figure 1 jpm-12-00801-f001:**
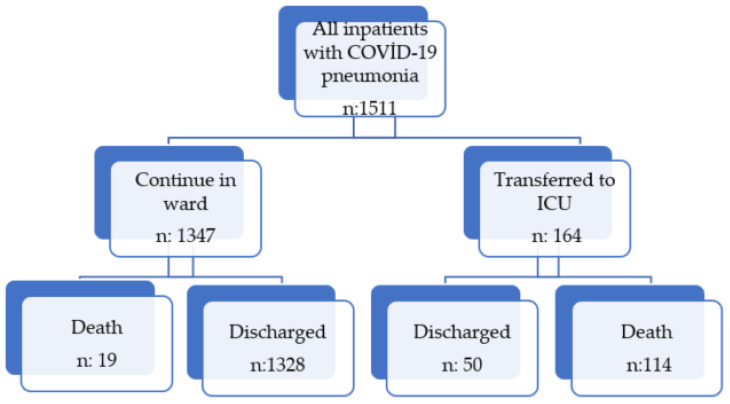
Flow chart of study population.

**Figure 2 jpm-12-00801-f002:**
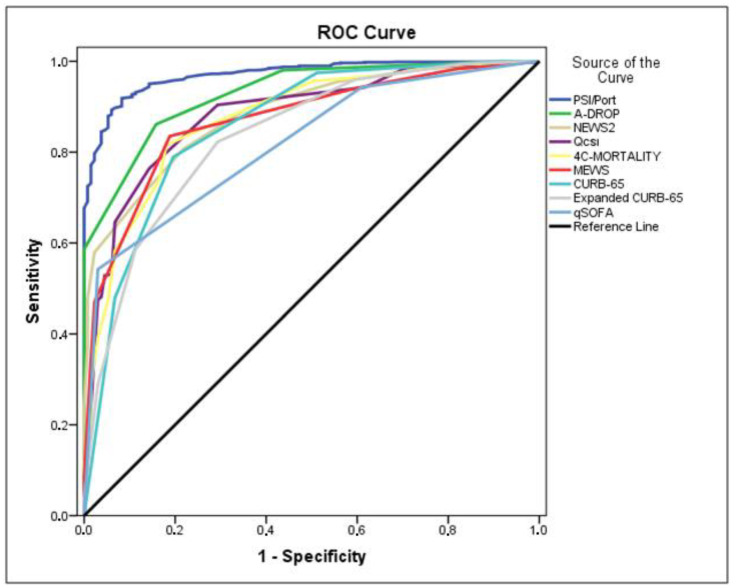
Receiver operating characteristic curves for PSI/PORT, A-DROP, NEWS2, qCSI, 4C-Mortality, MEWS, CURB-65, expanded CURB-65, and qSOFA scores for in-hospital mortality in COVID-19 pneumonia patients.

**Table 1 jpm-12-00801-t001:** The baseline characteristics of the patients.

	Non-Survivor Group (*n*: 133)	Survivors (*n*: 1378)	*p*
Age, years	72.8 ± 11.8	59.9 ± 14.7	<0.001
Male sex, *n* (%) 879 (58.17)	80 (60.15)	799 (58)	NS
Comorbidity no, %			<0.001
0	12 (9)	420 (30.47)	
1 (386, 25.54)	17 (12.78)	369 (26.77)	
≥2 (693, 45.86)	104 (78.19)	589 (42.74)	
Comorbidities, *n* (%)			
Hypertension, 726 (48.04)	92 (69.6)	634 (46.2)	<0.001
Diabetes, 503 (33.28)	52 (39.3)	451(32.9)	NS
Coronary artery disease, 217 (14.36)	41 (31)	176 (12.8)	<0.001
Atrial fibrillation, 85 (5.62)	25 (18.79)	60 (4.35)	<0.001
Congestive heart failure, 91 (6.02)	22 (16.54)	69 (5)	<0.001
Dyslipidemia, 73 (4.83)	8 (6.01)	65 (4.71)	NS
Cerebrovascular disease, 53 (3.5)	14 (10.6)	39 (2.8)	<0.001
Chronic obstructive pulmonary disease, 62 (4.1)	9 (6.8)	53 (3.8)	NS
Asthma, 135 (8.93)	12 (9)	123 (8.9)	NS
Malignancy, 75 (4.96)	13 (9.77)	62 (4.49)	0.01
Chronic kidney disease, 67 (4.43)	12 (9)	55 (4)	0.01
Physical findings			
Body temperature, °C	36.95 ± 0.64	36.91 ± 0.67	NS
Respiratory rate, per minute	30.26 ± 4.64	20.08 ± 4.34	<0.001
SpO_2_, under oxygen support, mean	92.95 ± 2.19	94.46 ± 1.90	<0.001
O_2_ support, L/per min	15.77 ± 9.94	3.86 ± 5.88	<0.001
Systolic blood pressure, mmHg	129.50 ± 22.28	126.51 ± 18.23	NS
Diastolic blood pressure, mmHg	70.14 ± 12.29	70.69 ± 10.28	NS
Heart rate, per minute	86.18 ± 20.58	82.68 ± 14.23	0.01

**Table 2 jpm-12-00801-t002:** Laboratory findings, CT scores, disease severity status and outcomes of the patients.

	Non-Survivor Group (*n*: 133)	Survivors (*n*: 1378)	*p*
Laboratory findings			
Neutrophil count, cells/mL	6.87 ± 3.76	5.40 ± 2.89	<0.001
Lymphocytes count, cells/mL	0.83 ± 0.56	1.21 ± 0.58	<0.001
N/L ratio (neutrophil/lymphocytes)	11.52 ± 9.99	5.74 ± 5.05	<0.001
Platelet count, 10^3^/mm^3^	215.30 ± 103.60	250.74 ± 105.14	<0.001
Hematocrit, %	36.70 ± 5.58	37.52 ± 4.72	NS
Glucose, mg/dL	171.91 ± 73.92	151.46 ± 71.50	0.003
Urea, mg/dL	71.80 ± 48.31	39.94 ± 25.62	<0.001
Creatinine, mg/dL	1.43 ± 1.54	0.94 ± 0.80	<0.001
Alanine transaminase, ALT, U/L	35.17 ± 30.87	43.66 ± 39.94	0.01
Aspartate aminotransferase, AST, U/L	49.42 ± 35.30	42.82 ± 30.90	0.04
Lactate dehydrogenase, LDH, U/L	480.73 ± 226. 37	344.84 ± 155.73	<0.001
Potassium, mEq/L	4.27 ± 0.60	4.22 ± 0.51	NS
Sodium, mEq/L	136.83 ± 5.70	137.19 ± 3.81	NS
C-reactive protein, mg/L	145.04 ± 78.54	101.63 ± 77.11	<0.001
Procalcitonin, ng/mL	0.79 ± 2.28	0.69 ± 7.87	NS
Ferritin, (µg/L)	770.39 ± 693.89	502.60 ± 562.91	<0.001
D-dimer, (µg FEU/mL)	1.32 ± 1.33	0.84 ± 1.21	<0.001
Fibrinogen, mg/dL	545.66 ± 146.22	511.43 ± 134.19	0.01
International normalized ratio, INR	1.16 ± 0.29	1.06 ± 0.20	<0.001
Troponin I, ng/mL	123.63 ± 449.44	18.49 ± 119.07	<0.001
Albumin, g/dL	32.78 ± 5.10	35.93 ± 5.23	<0.001
Disease Severity Status *n* (%)			<0.001
Moderate, 596 (39.44)	3 (2.25)	593(43.03)	
Severe, 915 (60.56)	130 (97.75)	785 (56.97)	
CT involvement *n*, (%)			<0.001
Mild, 329 (21.77)	13 (9.77)	316 (22.93)	
Moderate, 727 (48.11)	44 (33.8)	683 (49.56)	
Severe, 455 (30.11)	76 (56.43)	379 (27.50)	
Outcomes, *n* (%)			
Hospital length of stay, days	14.52 ± 8.78	11.27 ± 6.50	<0.001
Admission to ICU, 164 (10.85)	114 (85.71)	50 (3.62)	<0.001

**Table 3 jpm-12-00801-t003:** Prognostic scores of patients.

Score, Mean ± SD		Non-Survivor Group	Survivors	*p*
CURB-65	Mean ± SD	2.33 ± 1.05	0.75 ± 0.84	<0.001
	Median (Q1–Q3)	2 (2–3)	1 (0–1)	
Expanded CURB-65	Mean ± SD	4.18 ± 1.36	2.34 ± 1.85	<0.001
	Median (Min–Max)	4 (1–7)	2 (0–6)	
A-DROP	Mean ± SD	2.56 ± 0.94	0.57 ± 0.78	<0.001
	Median (Q1–Q3)	3 (2–3)	0 (0–1)	
qSOFA	Mean ± SD	1.41 ± 0.61	0.52 ± 0.61	<0.001
	Median (Q1–Q3)	1 (1–2)	0 (0–1)	
qCSI	Mean ± SD	7.02 ± 2.03	2.75 ± 2.90	<0.001
	Median (Q1–Q3)	7 (6–9)	2 (0–5)	
PSI/PORT	Mean ± SD	144.38 ± 28.64	67.17 ± 25.63	<0.001
	Median (Q1–Q3)	145 (124–168)	62 (49–81)	
NEWS2	Mean ± SD	8.29 ± 2.21	3.92 ± 2.71	<0.001
	Median (Q1–Q3)	8 (7–10)	4 (2–6)	
MEWS	Mean ± SD	3.44 ± 1.20	1.69 ± 1.05	<0.001
	Median (Q1–Q3)	3 (3–4)	2 (1–2)	
4C Mortality	Mean ± SD	13.96 ± 3.45	7.77 ± 3.99	<0.001
	Median (Min–Max)	14 (3–20)	8 (0–21)	

**Table 4 jpm-12-00801-t004:** The rates of in-hospital fatality and ICU admission across risk groups based on scores.

Risk Scores	No of Patients *n* (%)	Death *n* (%)	ICU Admission *n* (%)	Death in ICU *n* (%)
CURB-65				
0–1	1114 (73.72)	26 (1.72)	53 (3.5)	21 (1.38)
≥2	397 (26.27)	107 (7)	111 (7.34)	93 (6.16)
≥3	100 (6.61)	65 (4.3)	66 (4.36)	60 (3.97)
4	17 (1.12)	15 (0.99)	15 (0.99)	14 (0.92)
EXPANDED CURB-65				
0–1	402 (26.6)	4 (0.26)	8 (0.52)	4 (0.26)
≥2	1109 (73.09)	129 (8.53)	156 (10.32)	110 (7.27)
≥3	689 (45.59)	118 (7.8)	138 (9.13)	102 (6.75)
≥4	338 (22.36)	94 (6.22)	102 (6.75)	81 (5.36)
A-DROP				
0–1	1208 (79.94)	21 (1.38)	54 (3.57)	20 (1.32)
≥2	303 (20.05)	112 (7.41)	110 (7.27)	94 (6.22)
≥3	103 (6.81)	75 (4.96)	70 (4.63)	65 (4.3)
PSI/PORT				
≤70 (CLASS II)	872 (57.71)	0	8 (0.53)	0
71–90 (CLASS III)	282 (18.66)	5 (0.33)	12 (0.8)	2 (0.13)
91–130 (CLASS IV)	242 (16)	42 (2.77)	57 (3.77)	36 (2.38)
>130 (CLASS V)	115 (7.61)	86 (5.69)	87 (5.75)	76 (5)
≥107	341 (22.56)	122 (8.07)	132 (8.73)	108 (7.14)
MEWS				
0–2	1175 (77.76)	25 (1.65)	42 (2.77)	17 (1.12)
3	336 (22.23)	108 (7.14)	122 (8.07)	97 (6.41)
3–4	289 (19.12)	84 (5.55)	95 (6.28)	74 (4.89)
≥5	47 (3.11)	24 (1.58)	27 (1.78)	23 (1.52)
NEWS2				
0–4	801 (53)	3 (0.19)	12 (0.79)	2 (0.13)
5–6	339 (22.43)	26 (1.72)	30 (1.98)	20 (1.32)
≥6	547 (36.2)	118 (7.8)	136 (9)	101 (6.68)
≥7	371 (24.55)	104 (6.88)	122 (8)	92 (6)
qCSI				
≤3	741 (49)	8 (0.52)	13 (0.86)	4 (0.26)
4–6	544 (36)	31 (2)	37 (2.44)	24 (1.58)
≥6	439 (29)	114 (7.54)	135 (8.93)	103 (6.81)
7–9	222 (14.69)	93 (6.15)	110 (7.27)	85 (5.6)
10–12	4 (0.26)	1 (0.06)	4 (0.26)	1 (0.06)
4C MORTALITY				
0–3	223 (14.75)	1 (0.06)	2 (0.13)	0
4–8	589 (38.98)	8 (0.53)	20 (1.32)	7 (0.46)
9–14	573 (37.92)	49 (3.2)	79 (5.22)	49 (3.2)
≥15	126 (8.33)	66 (4.36)	63 (4.16)	58 (3.83)
≥12	363 (24)	109 (7.21)	112 (7.41)	94 (6.22)
qSOFA				
0	751 (49.7)	4 (0.26)	14 (0.92)	2 (0.13)
1	629 (41.6)	77 (5)	96 (6.35)	67 (4.43)
2	117 (7.74)	46 (3)	47 (3.11)	40 (82.64)
3	14 (0.92)	6 (0.39)	7 (0.46)	5 (0.33)

**Table 5 jpm-12-00801-t005:** Discriminative accuracy of scores in predicting hospital mortality.

Scores	AUROC (95% CI)	Std. Error	Cutoff	Se (%)	Sp (%)	PPV	NPV	*p*
PSI/PORT	0.971 (0.961–0.981)	0.005	≥107	91.7	91.9	52,1	99.1	<0.001
A-DROP	0.929 (0.911–0.948)	0.009	≥2	84.2	86.1	37.0	98.3	<0.001
NEWS2	0.885 (0.860–0.909)	0.012	≥7	78.2	80.6	28.0	97.5	<0.001
qCSI	0.882 (0.853–0.911)	0.015	≥6	85.7	76.4	26.0	98.2	<0.001
4C-MORTALITY	0.875 (0.845–0.906)	0.016	≥12	81.9	81.6	30.0	97.9	<0.001
MEWS	0.870 (0.842–0.898)	0.014	≥3	81.2	83.5	32.1	97.9	<0.001
CURB-65	0.859 (0.823–0.896)	0.019	≥2	80.5	78.9	27.0	97.7	<0.001
EXPANDED CURB-65	0.836 (0.800–0.873)	0.018	≥4	70.7	82.3	27.8	96.7	<0.001
qSOFA	0.818 (0.786–0.850)	0.016	≥1	97.0	54.2	17.0	99.5	<0.001

Abbreviations: AUROC: area under the receiver operating characteristic; CI: confidence interval; std. error: standard error; Se: Sensitivity; Sp: Specificity; NPV, negative predictive value; PPV, positive predictive value.

**Table 6 jpm-12-00801-t006:** Multivariate logistic regression analysis for risk factors on mortality.

	*p*	Odds Ratio	%95 CI	
			Lower	Upper
PSI/PORT (≥107)	0.001 **	25.172	11.232	56.413
A-DROP (≥2)	0.001 **	4.686	2.303	9.532
MEWS (≥3)	0.009 **	2.458	1.255	4.814
qSOFA (≥1)	0.003 **	5.714	1.774	18.399
O_2_ support, L/per min	0.001 **	1.065	1.027	1.105
Platelet count, PLT, 10^3^/mm^3^	0.024 *	0.997	0.995	1.000
C-reactive protein, CRP, mg/L	0.046 *	0.996	0.992	1.000
Lactate dehydrogenase, LDH, U/L	0.002 **	1.003	1.001	1.004
Constant	0.001 **	0.001		

* *p* < 0.05; ** *p* < 0.01.

## Data Availability

The data presented in this study are available on request from the corresponding author. The data are not publicly available due to privacy reasons.

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
