# Peer review of "Comparison of Pneumonia Severity Indices, qCSI, 4C-Mortality Score and qSOFA in Predicting Mortality in Hospitalized Patients with COVID-19 Pneumonia"

_jpm, 2022, doi:10.3390/jpm12050801_

Round 1
Reviewer 1 Report
Thankyou for asking me to review this interesting paper on pneumonia scoring systems in predicting outcomes in COVID-19 pneumonitis. This large retrospective cohort study provides information on the merits of various pneumonia scoring systems in predicting death and disease progression in patients with COVID-19. The authors also compare several other known variables that are different between survivors and non-survivors. The authors suggest that the PSI port and A drop were the most sensitive and specific scores in predicting hospital outcome for patients with COVID-19.
This is a large retrospective study in an area of interest. The authors make appropriate reference to other papers looking at similar scoring systems. Whilst this increases the validity of the findings it does also rather detract from this manuscript’s novelty. The methods are clear and well described. The tables are clear but rather long and could either be split up or placed in an online repository. The main concern about this study is the lack of control for multiple comparisons which does not seem to have been done and may significantly alter the results. Further statistical advice may increase the impact of the data this paper presents. The confidence intervals for the AUC ROC curves for PSI PORT and A-DROP show that they were superior to other predictive models which is interesting. The clinical relevance for patients with high scores is not clear. The authors conclusions about using A-DROP or even the qCSI which were inferior tests according to the data is unsubstantiated and would need further explanation possibly with results showing the time taken to fill out these assessments.
The authors should:
Consider splitting up the tables to make them more digestible in a single glance
The method by which control for multiple comparisons was carried out should be described
Further statistical advice should be taken on this and the subsequent results and details of statistician involvement should be included in the methods
The superiority of the PSI/PORT and ADROP should be emphasised in the discussion
The discussion should be expanded with comments on the relative merits of each questionnaire
A section on the clinical relevance of high scores should be included in the discussion
Limitations and strengths section should be added to the discussion
The conclusion about which measure to use should be modified or expanded to better reflect the rest of the manuscript
Minor points
Abstract
Line 29 existence – not sure what this means
Line 31-32 sentence on age is redundant
Line 35 needs ROC curve data
Introduction
94 – Probably needs a reference. Do scoring systems really reduce mortality in ICU.
Methods
100 – Single or multi-centre?
Results
155 About?
158 -159 not sure this is needed
Table 1 - needs all the acronyms explained
- Hospital LOS median and IQR looks like mean and sd
Discussion
252 please define effective (lowest sensitivity 70%)
258 and statistically higher than other scores measured in this population (no overlap in CI)
271 Again the CI should be referred to here for clarity of what the data has shown
280 -281 (for example) The comparison to previous studies is numerical and should be treated with more caution. There is no statistical test showing this
309 The PSI/PORT advantages and disadvantages needs more discussion before leading to the subsequent conclusion
Author Response
RESPONSE TO REVIEWER 1
Thank you for evaluating our manuscript. A native speaker has reviewed and corrected the text.
The research design was based on the relevance of different SCORE models in predicting in-hospital mortality; nine score systems were included and compared.
Our main conclusions are that PSI-PORT and A-DROP are the most clinically relevant scores for predicting in-hospital mortality.
The tables are split up according to your recommendations.
Further statistical analysis was carried out.
The method by which control for multiple comparisons was carried out was described.
The superiority of the PSI/PORT and ADROP was emphasised in the discussion.
The discussion was expanded with comments on the relative merits of each questionnaire.
A section on the clinical relevance of high scores is included in the discussion.
Limitations section was added to the discussion.
The conclusion about which measure to use was expanded .
Minor points
Abstract
Line 29 existence – was removed.
Line 31-32 sentence on age is removed
Line 35 ROC curve data is given.
Introduction
94 – A reference is provided.
Methods
100 – (revised edition line 104) Single-centre
Results
155 (revised edition line 163) About is removed
158 -159 This was deleted.
Table 1 - All the acronyms are explained
Hospital LOS median and IQR were corrected as mean and sd +
Discussion
252 (revised edition line 343) Effective was changed as superior and defined.
258 (revised edition line 353) Was changed as you recommend.
271(revised edition line 391) The CI was given here for clarity of what the data has shown.
280 -281 (revised edition line 408-413) The comparison to previous studies is revised and presented in accordance.
309 (revised edition line 359) The PSI/PORT advantages and disadvantages were were more thoroughly discussed.
All revisons were pointed in red font.
Reviewer 2 Report
In this paper, authors investigated the comparison of pneumonia severity indices, qCSI, 4C-Mortality score and qSOFA in predicting mortality in hospitalized patients with COVID-19 pneumonia.
The authors included a relatively large number of patients in the analysis, but the disadvantage of the analysis is the retrospective nature of the study.
Another significant limitation of this study is the inclusion of patients from the early periods of the COVID-19 pandemic and the lack of assessment of the impact of new SARS-CoV2 variants (e.g. omicron) and vaccines on mortality. For this reason, the obtained results may be of limited clinical utility.
The results obtained from the study could have been quite clinically useful in the early stages of the COVID-19 pamdemia. However, the dynamically changing course of COVID-19 caused by successive variants of SARS-CoV2 has a rather limited usefulness of the results obtained.
In conclusion:
In my opinion, the manuscript is not eligible for publication. Significant limitation of this study is the inclusion of patients from the early periods of the COVID-19 pandemic and the lack of assessment of the impact of new SARS-CoV2 variants (e.g. omicron) and vaccines on course of disease and mortality. For this reason, the obtained results may be of limited clinical utility.
Author Response
Thank you for reviewing our manuscript.
The introduction was improved.
All the cited references were improved.
The research design was improved.
Further statistical analysis was provided.
The results were extended by splitting tables. The results were improved using new statistical data.
We accept the disadvantage of the analysis is the retrospective nature of the study, and inclusion of patients from the early periods of the COVID-19 pandemic and the lack of assessment of the impact of new SARS-CoV2 variants (e.g. omicron) and vaccines on mortality. The obtained results must be validated in new cohorts.
Round 2
Reviewer 1 Report
The manuscript has been significantly updated an improved.
Minor points - A mention of the APORT needs to be put into the abstract prior to concluding on its suitability.
Within the results are the OR really 25,172 or is it 25.172
A point about the multiple comparisons issue I cannot find that in the manuscript stats methods or limitations
Reviewer 2 Report
The authors have satisfactorily responded to all my questions and made the necessary changes to the manuscript.